# Influence of a change in activity regime on femoral bone architecture and failure behaviour

Claire C. Villette[1,2], Andrew T. M. Phillips[1,2]*

**1** Department of Civil and Environmental Engineering, Structural Biomechanics, Imperial College London, London, United Kingdom, **2** The Royal British Legion Centre for Blast Injury Studies, Imperial College London, London, United Kingdom

* andrew.phillips@imperial.ac.uk

## Abstract

The incidence and morbidity of femoral fractures increases drastically with age. Femoral architecture and associated fracture risk are strongly influenced by loading during physical activities and it has been shown that the rate of loss of bone mineral density is significantly lower for active individuals than inactive. The objective of this work is to evaluate the impact of a cessation of some physical activities on elderly femoral structure and fracture behaviour. The authors previously established a biofidelic finite element model of the femur considered as a structure optimised to loading associated with daily activities. The same structural optimisation algorithm was used here to quantify the changes in bone architecture following cessation of stair climbing and sit-to-stand. Side fall fracture simulations were run on the adapted bone structures using a damage elasticity formulation. Total cortical and trabecular bone volume and failure load reduced in all cases of activity cessation. Bone loss distribution was strongly heterogeneous, with some locations even showing increased bone volume. This work suggests that maintaining the physical activities involved in the daily routine of a young healthy adult would help reduce the risk of femoral fracture in the elderly population by preventing bone loss.

## Introduction

The incidence and morbidity of femoral fractures increases drastically with age [1, 2]. Three factors are commonly listed as contributors to the risk of fracture: bone quality, the risk of falling (including physical conditions and environmental situations that increase the likelihood of falling), and the effectiveness of neuromuscular response that protects the skeleton from injury [2, 3]. All three factors are aggravated with increasing age [3–5]. Bone mineral density (BMD) decreases with age, and the rate of loss of BMD can reach up to 2% per year for osteoporotic patients [5, 6]. Many fracture prevention programs among the elderly focus on reducing the risk of falling [3, 4]. This involves tackling some of the predisposing environmental conditions, for example via 'fall-proofing' one's home [3]. Elderly individuals might of their own accord reduce or stop performing fall-inducing activities such as stair climbing due to apprehension

**Data Availability Statement:** All relevant data are within the paper and its Supporting information files.

**Funding:** CV was funded by the Royal British Legion Centre for Blast Injury Studies at Imperial

College London. The funders had no role in study design, data collection and analysis, decision to publish, or preparation of the manuscript.

**Competing interests:** The authors have declared that no competing interests exist.

or physical weakness. For instance, in the United States, the National Health Interview Survey reported a substantial decline in regular, light to moderate activity for both sexes after 74 years old [7]. However, it is commonly accepted that bone strength is improved by physical activity [5]. It has been shown that the rate of loss of BMD is significantly lower for active individuals than inactive [5, 8, 9]. For this reason, it is of interest to investigate how maintaining one's daily physical activity routine might protect bone integrity. The authors have previously studied the adaptation of human bones [10–16] to activities representative of daily loading [17] including walking, stair climbing, sit-to-stand and light weight lifting. This study investigates predicted changes in the femur structure, as well as their influence on fracture load and pattern, as some activities stop being performed.

## Materials and methods

### Femur adaptation to a new activity regime

The bone remodelling algorithm described in Phillips et al. [12] was used for this investigation. A brief summary of its logic is given in this paragraph and the reader is invited to refer to the original publication for a more detailed description. The femur finite element (FE) model is made of shell elements representing cortical bone and truss elements representing trabecular bone. As the model is submitted to a series of loading cases, the absolute maximum strain $\epsilon_{max}$ is defined as the absolute maximum axial strain in the truss elements and as the absolute maximum in-plane principal strain in the internal surface for the shell elements, across all loading scenarios. Based on the Mechanostat hypothesis [18] which defines a target strain $\epsilon_t$ for bone, the radius $r$ and the thickness $t$ of every truss and shell element, respectively, is adapted linearly with respect to the ratio $\epsilon_{max} : \epsilon_t$ The adapted FE model is then iteratively submitted to the same loading and adaptation steps until convergence of the structure.

In Phillips et al. [12], a biofidelic femur model was obtained using this remodelling algorithm under a combination of loading scenarios representative of the daily living activities [19]: walking (11 different loading scenarios corresponding to 11 frames of the gait cycle), sit-to-stand (20 frames), stand-to-sit (22 frames), stair ascent (12 frames) and stair descent (15 frames). This selection of activities was taken as representative of the normal activity regime of a healthy adult. This loading scenario was used to generate a reference femur model referred to as the original femur model.

Three changes to the activity regime of the healthy adult were introduced. For the first one, a cessation of stair ascent and stair descent activities was simulated while maintaining sit-to-stand, stand-to-sit, and walking activities. For the second, a cessation of sit-to-stand and stand-to-sit activity was modelled, while maintaining stair ascent, stair descent, and walking activity. The third change modelled a cessation of stair ascent, stair descent, sit-to-stand, and stand-to-sit activities, leaving walking as the only activity performed. The remodelling algorithm was run from the original femur model under the loading scenarios corresponding to the remaining activities in all three cases. The femur models obtained after convergence of the optimisation for cases one, two and three were named respectively $Femur_{NoStairs}$, $Femur_{NoSit}$, and $Femur_{WalkOnly}$.

### Assessment of bone structural changes

The volumes of trabecular $V_{Trabecular}$ and cortical bone $V_{Cortical}$, and the total bone volume $V_{Total}$ in the original femur and in $Femur_{NoStairs}$, $Femur_{NoSit}$, and $Femur_{WalkOnly}$ were

calculated as:

$$V_{\text{Trabecular}} = \sum_j \pi r_j^2 l_j \tag{1}$$

$$V_{\text{Cortical}} = \sum_j f_j t_j \tag{2}$$

$$V_{\text{Total}} = V_{\text{Trabecular}} + V_{\text{Cortical}} \tag{3}$$

with subscript $j$ referring to the $j^{th}$ truss (trabecular bone) or shell (cortical bone) element, $r$; the truss radius, $l$; the truss length, $f$; the area of the shell face, and $t$; the shell thickness. The percentage difference $\Delta V_{\text{Trabecular}}$, $\Delta V_{\text{Cortical}}$, $\Delta V_{\text{Total}}$ in each of these three quantities between the original femur model and the converged models in response to the three new activity regimes were calculated, as well as the percentage differences in truss element cross-sectional area $\Delta A_j$ and shell thickness $\Delta t_j$ for each truss and shell elements, with $A_j = \pi r_j^2$.

## Fracture simulations in side fall scenario

The influence of the changes in activity regime on failure risks, including failure load and fracture pattern, was assessed using the scenario of lateral compression implemented in Villette and Phillips [19] to represent a fall to the side, with the damage elasticity model developed as an *Abaqus* subroutine released as supplementary material to Villette and Phillips [19]. In a few words, this scenario models a displacement driven compression in a direction close to perpendicular to the shaft axis, applied via a loading plate pressing onto the greater trochanter while the femoral head is constrained by contact with a fixed stiff plate. At each time step, the strains at each section point of the elements are compared to compressive and tensile yield and ultimate strain criteria. Truss elements representing trabecular bone in the adapted femur models, are converted to beam elements in the fracture simulations. Yield strain (indicated with a $y$ subscript) in compression ($c$ subscript) and tension ($t$ subscript), and ultimate strains ($u$ subscript) in tension and compression were set to values: $\epsilon_{y,c} = -0.0084$, $\epsilon_{y,t} = 0.0060$, $\epsilon_{u,c} = -0.0116$ and $\epsilon_{u,t} = 0.0190$, respectively [19]. If one of these criteria is reached, the corresponding section point is assigned a reduced Young's modulus, based on the damage elasticity material model defined for bone. Upon reaching the ultimate strain criteria, the section point is assigned a Young's modulus three orders of magnitude lower than the Young's modulus of bone assigned initially to all the elements in the model (E = 18.0 GPa), which effectively cancels its load bearing ability. The yielded and failed elements are identified, allowing visualization of the fracture pattern. The failure load is taken as the maximum reaction force recorded at the fixed plate nodes over the simulation.

## Results

### Structural bone changes

The total bone volume in the original femur amounted to 204.2 cm$^3$, distributed between around 60% of cortical bone and 40% of trabecular bone. Fig 1 presents the percentage difference in bone volume as the original femur model is adapted to the reduced activity regimes. All three new activity regimes result in a decrease in total bone volume with $\Delta V_{\text{Total,NoStairs}} = -4.1\%$, $\Delta V_{\text{Total,NoSit}} = -5.5\%$, and $\Delta V_{\text{Total,WalkOnly}} = -12.0\%$. A marked decrease in trabecular bone volume is observed with $\Delta V_{\text{Trabecular,NoStairs}} = -9.0\%$, $\Delta V_{\text{Trabecular,NoSit}} = -13.1\%$, and $\Delta V_{\text{Trabecular,WalkOnly}} = -23.7\%$. To a lesser extent, a decrease in cortical bone volume is also

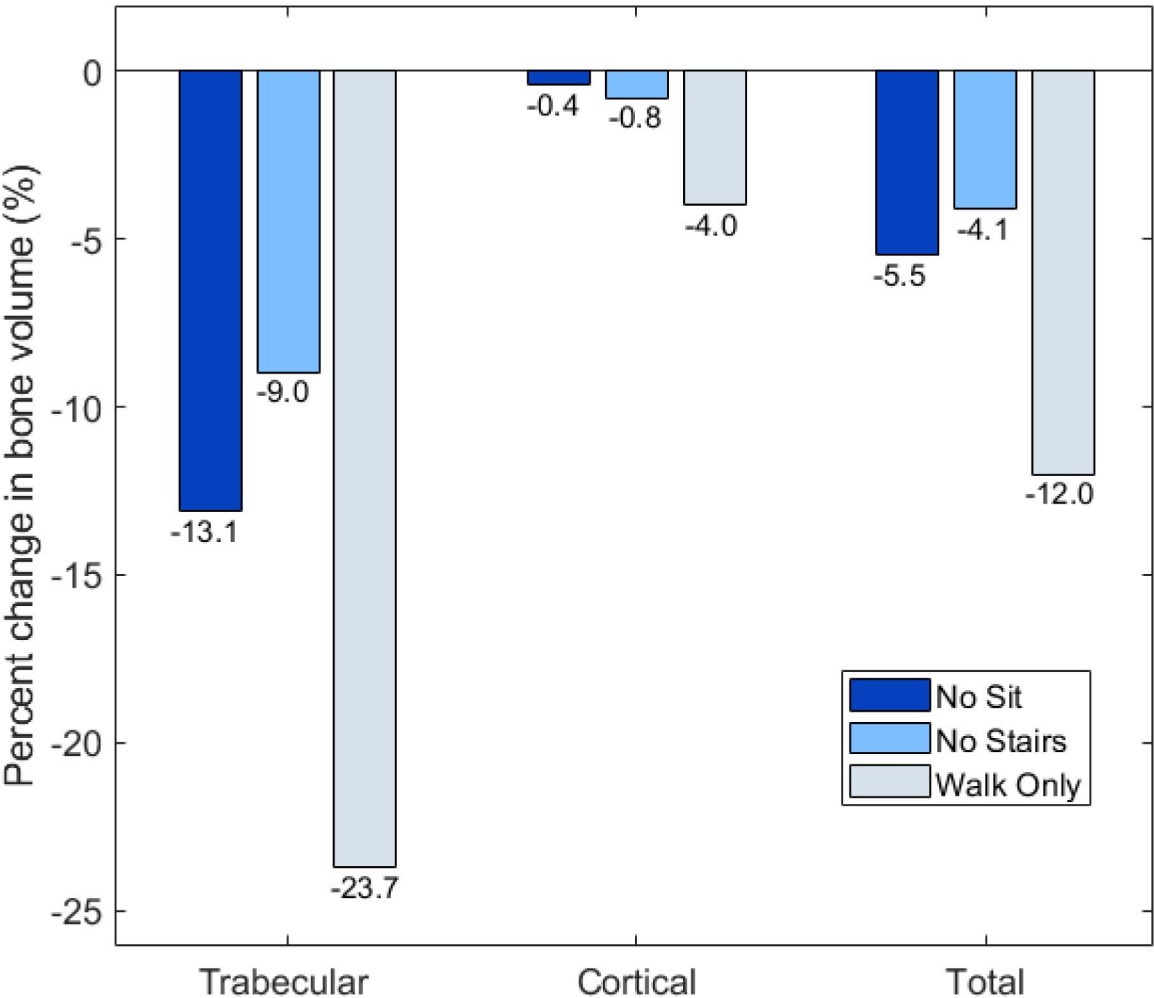

**Fig 1. Percent changes in bone volume $\Delta V_{\text{Trabecular}}$, $\Delta V_{\text{Cortical}}$, and $\Delta V_{\text{Total}}$ between the original femur model [12] and each of the lower intensity activity regimes: NoStairs, NoSit, and WalkOnly.**

observed, with $\Delta V_{\text{Cortical,NoStairs}} = -0.8\%$, $\Delta V_{\text{Cortical,NoSit}} = -0.4\%$, and $\Delta V_{\text{Cortical,WalkOnly}} = -4.0\%$.

Fig 2 describes the relative contribution of cortical and trabecular bone to the total bone loss associated with the reduced activity regimes.

Figs 3 to 5 show the location of the elements whose cross-sectional area has been changed in the reduced activity regimes. The cross-sectional area of an element $j$ is considered as changed if $|\Delta A_j| > 5\%$ or $|\Delta t_j| > 5\%$. A more detailed distribution of the values of $\Delta A$ and $\Delta t$ is presented in Tables 1 and 2. Femur$_{\text{NoStair}}$ (Fig 3) presents a reduced cortical thickness (representing bone resorption) in a large zone of the medial anterior distal half of the bone, including the distal diaphysis, metaphysis and epiphysis, as well as in the proximal posterior half of the shaft, including the proximal diaphysis and metaphysis. On the other hand, smaller zones of increased cortical thickness (representing bone apposition) are observed in the proximal anterior part of the bone as well as in the distal posterior part. Small zones of reduced cortical thickness are also observed on the superior greater trochanter, the superior anterior neck and

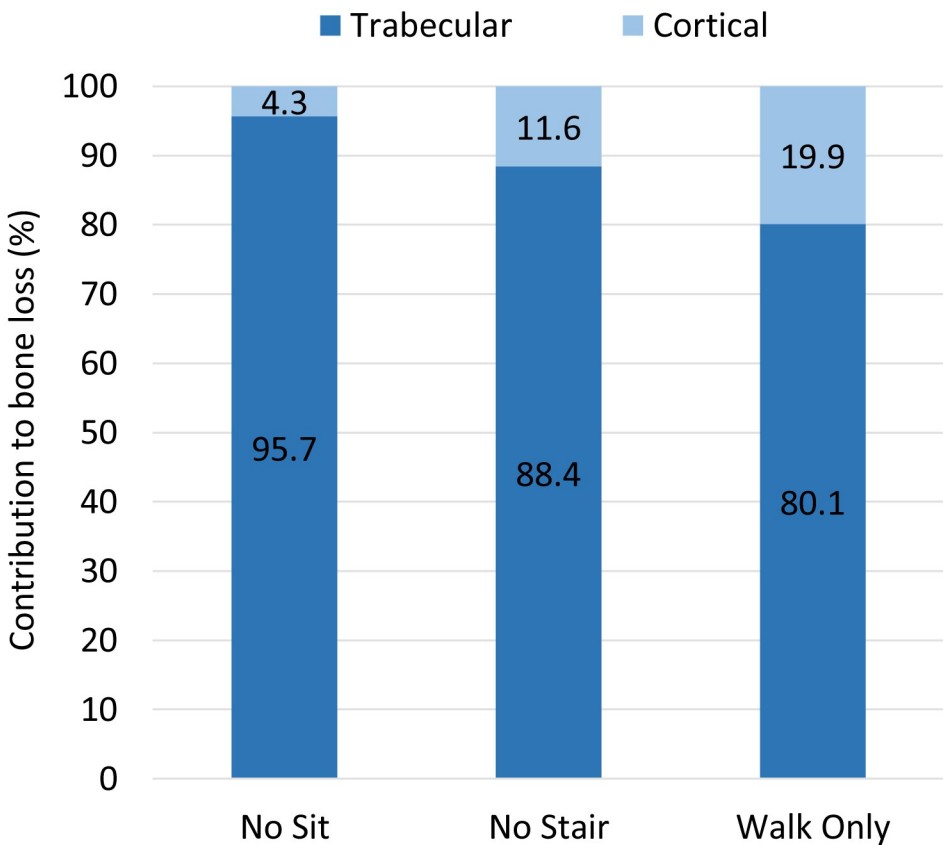

**Fig 2. Relative contribution of cortical and trabecular bone to the total bone loss associated with the reduced activity regimes.**

the superior posterior head. Referring to the main trabecular groups identified by Singh et al. [20], some reduction in cross-sectional area (representing bone resorption) is observed in the secondary tensile group as well as in the greater trochanter group, while some increase (representing bone apposition) is observed in the primary compressive, secondary compressive, and primary tensile groups. The distal trabecular bone shows some resorption in the medial side, and some apposition in the lateral side.

Femur$_{NoSit}$ (Fig 4) shows a reduced cortical thickness in a zone of the distal lateral posterior diaphysis (shaft) as well as in a thin band on the proximal lateral anterior part of the bone. Some increase in cortical thickness is observed in the distal medial posterior and distal lateral anterior parts of the bone. Some isolated shells also present an increased thickness in the proximal anterior part of the bone as well as along the diaphysis. Bone resorption is observed in the greater trochanter and in the neck region, while an increase in trabecula cross-sectional areas is apparent in the secondary compressive and primary tensile groups. Most of the distal lateral trabecular bone shows bone resorption, with some elements close to the bone longitudinal axis exhibiting an increase in cross-sectional area.

Femur$_{WalkOnly}$ (Fig 5) presents two bands of reduced cortical thickness, on the anterior aspect, from the superior neck to the distal medial anterior part of the bone, and on the posterior aspect, from the proximal medial extremity of the diaphysis to its distal lateral extremity,

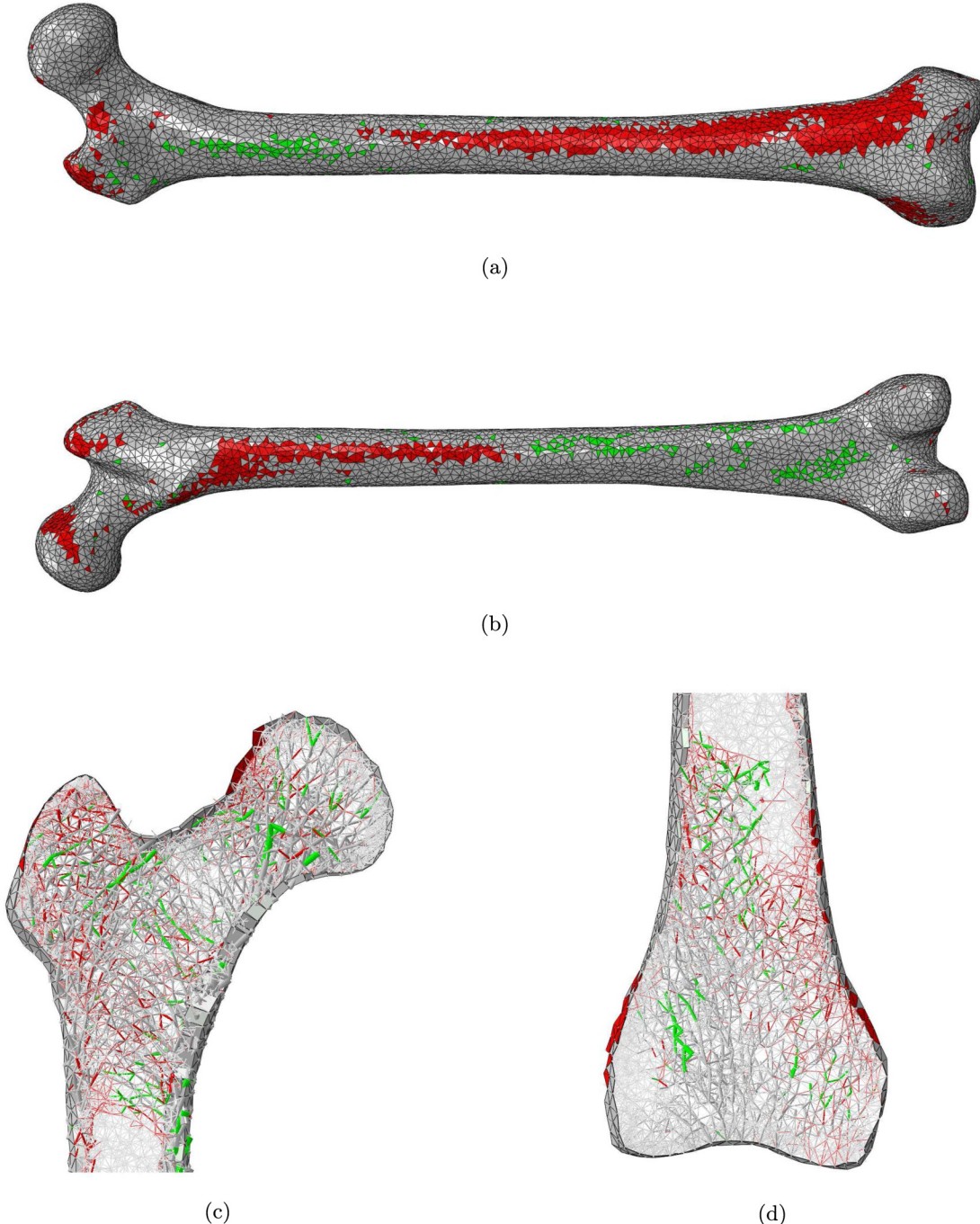

**Fig 3. a) anterior and b) posterior views of the femur cortex and longitudinal slices of c) the proximal and d) the distal epiphysis of the Femur$_{NoStair}$ model.** Elements with a decreased or increased cross-section are displayed in red and green respectively.

likely due to a reduction in the range of flexion and extension at the hip and knee joints with cessation of stair ascent and descent, sit-to-stand, and stand-to-sit, resulting in a reduced bending moment envelop acting on the femur in the sagittal plane, in comparison to the full range of daily loading activities. Smaller zones of increased cortical thickness are observed on

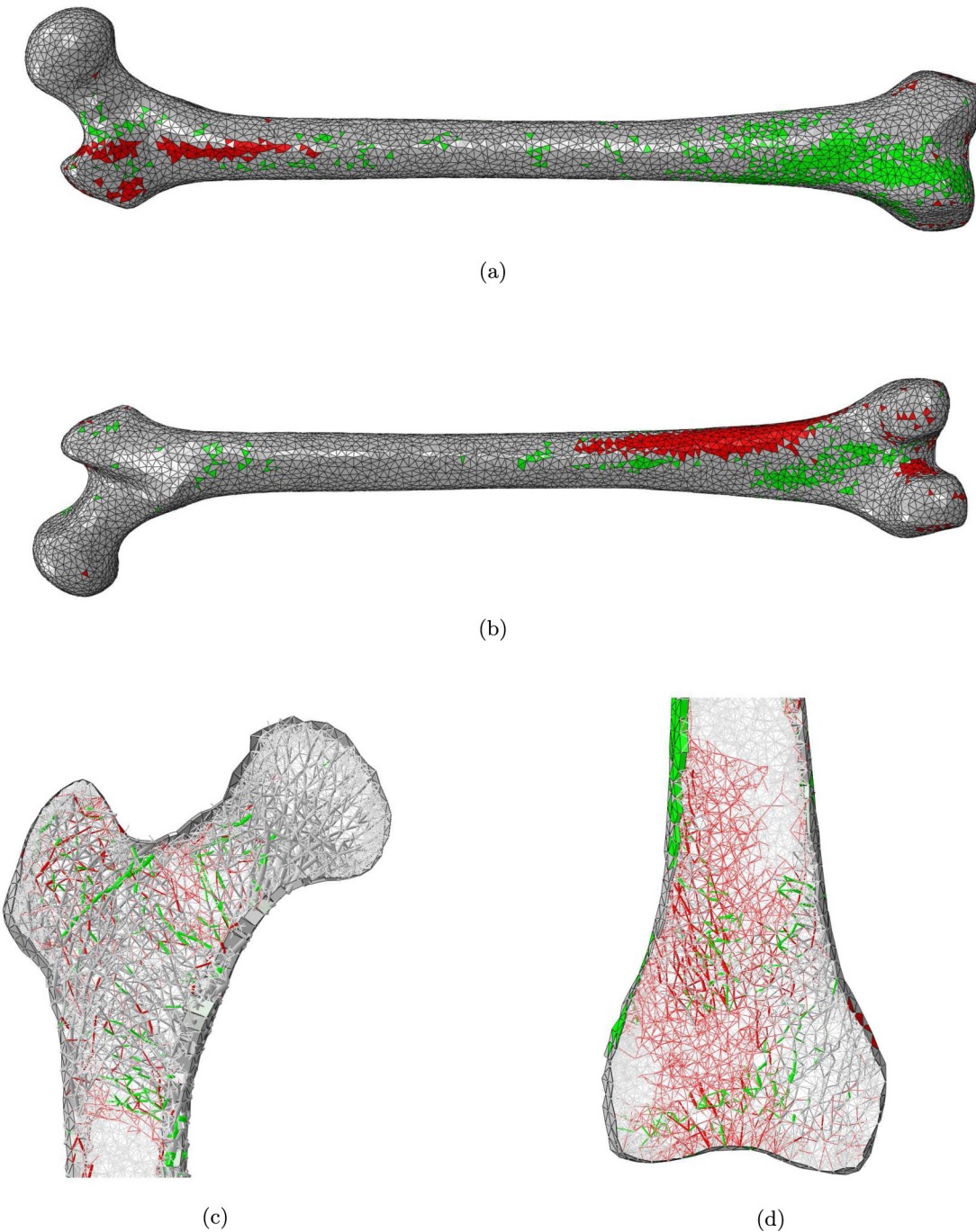

**Fig 4. a) anterior and b) posterior views of the femur cortex and longitudinal slices of c) the proximal and d) the distal epiphysis of the Femur$_{NoSit}$ model.** Elements with a decreased or increased cross-section are displayed in red and green respectively.

the distal lateral anterior part and the distal posterior extremity of the diaphysis. Marked zones of bone apposition are observed in the proximal trabecular bone, respectively in the primary and secondary compressive groups as well as in the primary tensile group. Bone resorption is marked in the neck, the greater trochanter, and the secondary tensile group. Almost the

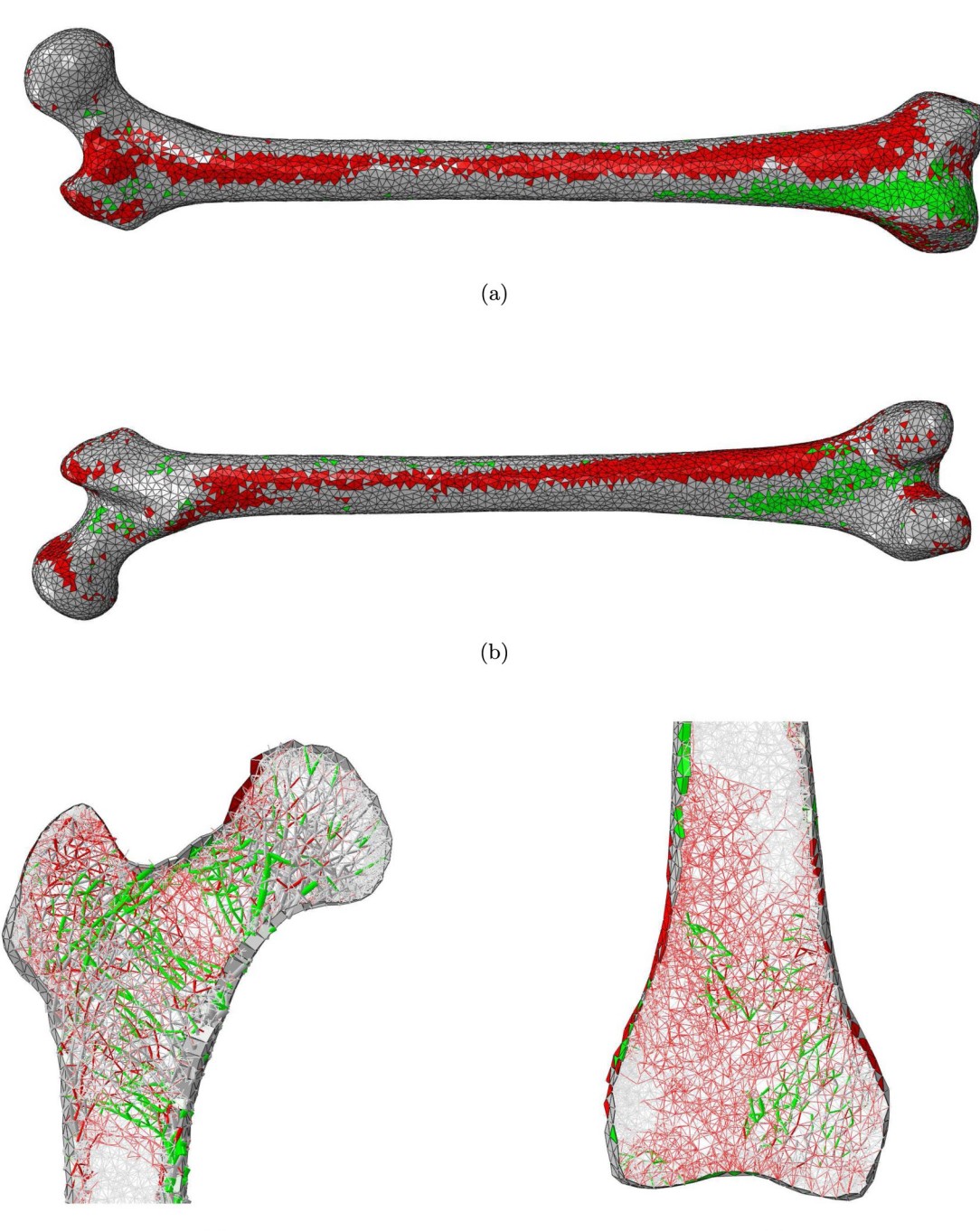

**Fig 5. a) anterior and b) posterior views of the femur cortex and longitudinal slices of c) the proximal and d) the distal epiphysis of the Femur_WalkOnly model.** Elements with a decreased or increased cross-section are in red and green respectively.

**Table 1. Distribution (%) of the values of ΔA across the trabecular elements in the new activity regimes.** Bracketed intervals are left-closed.

| ΔA | <-90 | [-90,-50] | [-50,-10] | [-10,-5] | [-5,5] | [5, 10] | [10, 50] | [50, 90] | >90 |
|---|---|---|---|---|---|---|---|---|---|
| Femur_NoStairs | 6.98 | 0.41 | 3.74 | 0.16 | 87.81 | 0.21 | 0.58 | 0.06 | 0.03 |
| Femur_NoSit | 9.57 | 0.66 | 3.07 | 0.13 | 85.39 | 0.17 | 0.78 | 0.16 | 0.08 |
| Femur_WalkOnly | 20.46 | 0.78 | 4.30 | 0.18 | 72.12 | 0.34 | 1.31 | 0.30 | 0.20 |

**Table 2. Distribution (%) of the values of Δ*t* across the shell elements in the new activity regimes.** Bracketed intervals are left-closed.

| Δ*t* | <-90 | [-90,-50] | [-50,-10] | [-10,-5] | [-5,5] | [5, 10] | [10, 50] | [50, 90] | >90 |
|---|---|---|---|---|---|---|---|---|---|
| Femur$_{NoStairs}$ | 0.00 | 0.70 | 11.17 | 0.90 | 84.54 | 0.63 | 2.05 | 0.00 | 0.00 |
| Femur$_{NoSit}$ | 0.00 | 0.20 | 5.41 | 0.35 | 85.81 | 1.32 | 5.71 | 0.73 | 0.48 |
| Femur$_{WalkOnly}$ | 0.04 | 3.08 | 19.44 | 1.25 | 69.90 | 0.70 | 3.39 | 0.87 | 1.32 |

entirety of the distal trabecular bone shows resorption, with the exception of a patch of elements on the medial side and another one in the center.

## Impact of the structural changes on bone failure

The fracture patterns and failure loads for the original regime of daily loading activities and the three reduced activity regimes are displayed in Fig 6. The force-displacement structural responses are plotted in Fig 7. Failure of the original femur model was reached for a load F = 4.2 kN. Femur$_{NoStairs}$, Femur$_{NoSit}$, Femur$_{WalkOnly}$ reached structural failure at respective loads F = 3.9 kN, F = 3.8 kN, and F = 3.3 kN. The fracture pattern obtained for Femur$_{WalkOnly}$ contrasts with the others, with a greater trochanter quasi-complete separation and a fracture development down the frontal aspect of the shaft. Three potential fragments can be isolated, with loss of posterolateral support, which resembles a Type 3 fracture in Steen Jensen and Michaelsen's classification [21–23]. The fracture pattern obtained for Femur$_{NoStairs}$ is almost identical to that of the original femur, with an initial crack propagating from the superior-lateral neck down the greater trochanter, followed by an intertrochanteric second fracture line, forming three fragments with loss of medial support. This pattern suggests a Type 4 intertrochanteric fracture according to Steen Jensen and Michaelsen's classification [21–23]. The fracture pattern obtained for Femur$_{NoSit}$ is very similar to that of the original femur, although the initial crack propagation almost results in complete intertrochanteric fracture below the lesser trochanter, before the second intertrochanteric fracture line is created just above the lesser trochanter. More damage is also visible at the junction of the greater trochanter with the femoral neck. The force-displacement response for Femur$_{NoStairs}$ is initially very similar to that of the original femur, with an initial peak reached at 3.9 kN load. That peak corresponds to the initial crack appearance and is reached for a displacement of 1.5 mm for both Femur$_{NoStairs}$ and Femur$_{Original}$. However, while the original femur then maintains a positive stiffness and reaches a higher failure load, Femur$_{NoStairs}$ maintains this load level for some time before failing. The first peak observed for Femur$_{NoSit}$ is of lower amplitude, but the femur retains a positive stiffness and reaches its maximum reaction force for a 4 mm displacement. The first peak observed for Femur$_{WalkOnly}$ is lower still, and its higher reaction force is reached about 1.5 mm later.

## Discussion

The patterns of bone resorption and growth observed in Figs 3 and 4 are consistent with the observations discussed in Phillips et al. [12] regarding the most influential activities for the determination of shell and truss element cross-sectional areas. As an example, stair climbing activities are the most influential for the elements located in the superior greater trochanter and the proximal posterior cortex, which explains their resorption in Femur$_{NoStairs}$ when these activities are stopped. Similarly, sitting activities are the most influential for the elements located in the distal lateral femur and the distal posterior femur, which explains their resorption in Femur$_{NoSit}$ when these activities are stopped.

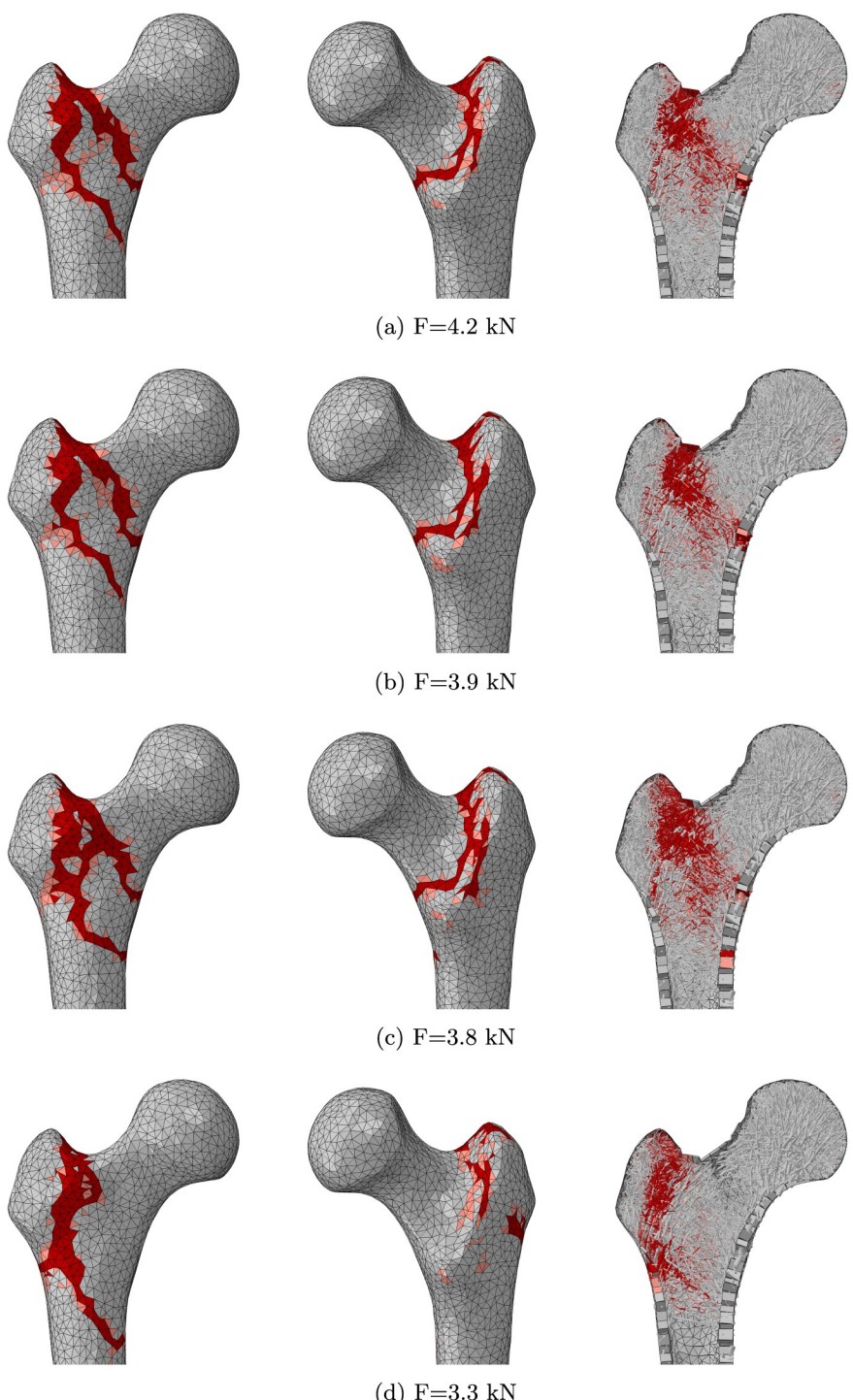

(a) F=4.2 kN

(b) F=3.9 kN

(c) F=3.8 kN

(d) F=3.3 kN

**Fig 6. Fracture patterns and failure loads in side fall simulations for a) the original femur model, b) Femur$_{NoStairs}$, c) Femur$_{NoSit}$ and d) Femur$_{WalkOnly}$.** Yielded and failed elements are displayed in pink and red respectively.

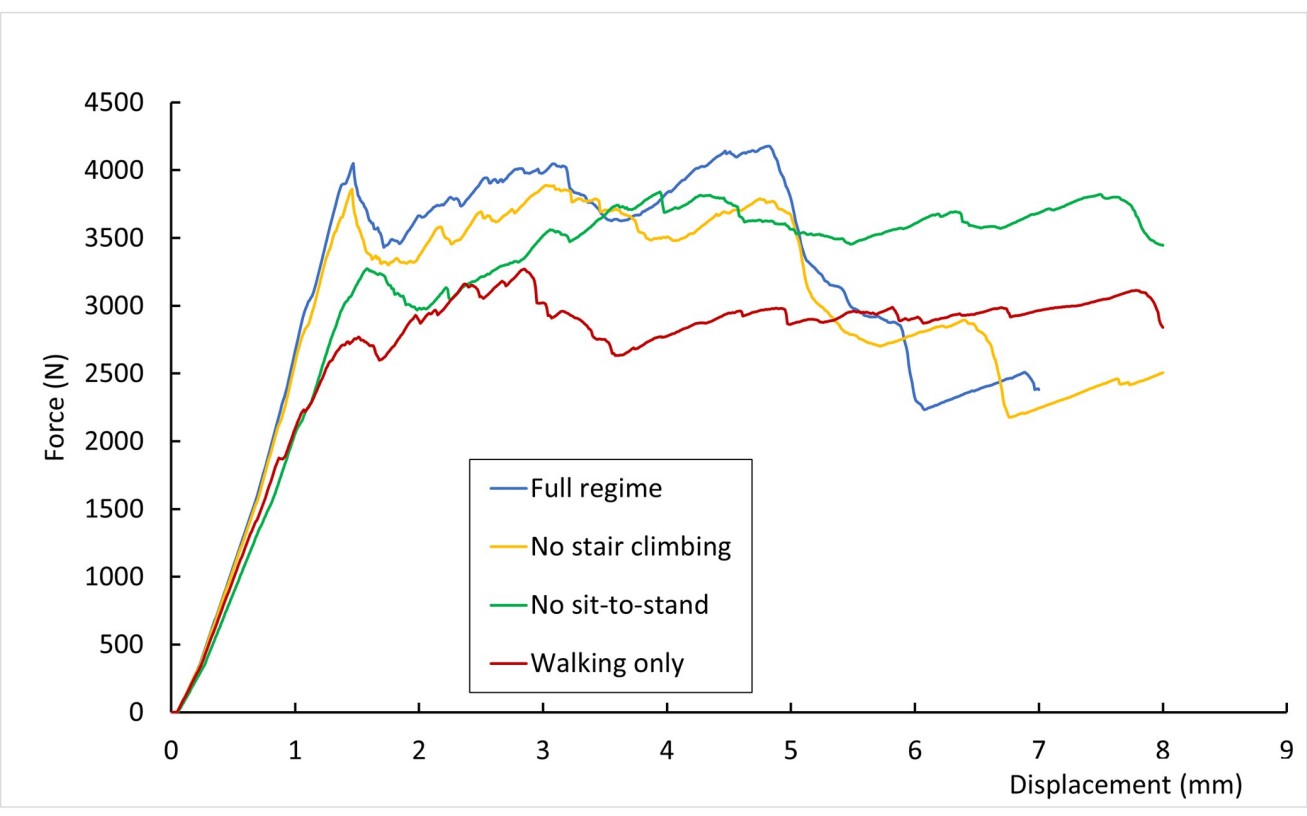

**Fig 7. Force-displacement responses during side fall simulations of the femurs optimised to different activity regimes: Full activitiy regime used for the original femur model, and reduced activity regimes.**

Tables 1 and 2 present the distribution of the values of $\Delta A$ and $\Delta t$ across all the elements in the new activity regimes. A vast majority of elements maintain a cross-sectional area similar to that of the original femur model (>69% in all cases, and >84% for $Femur_{NoStairs}$ and $Femur_{NoSit}$). As expected, more remodelling occurs in the regime corresponding to the most drastic change in activities, $Femur_{WalkOnly}$. In all three remodelling cases, trabecular bone exhibits mostly bone resorption (never more than 2.5% of the elements show an increase in cross-sectional area). For many of the resorbing trabecular elements, the cross-sectional area is reduced by more than 90%. While most of the structural changes in the cortex also involve bone resorption, most of the adapted shells present a thickness reduction of less then 50%, and very rarely greater than 90%. Interestingly, there are more shells growing than resorbing in $Femur_{NoSit}$. This is consistent with the idea of structural optimisation, involving the remodelling of the whole structure towards a more 'specialised' structure for the remaining loading cases. Indeed, as shown in Phillips et al. [12], sit-to-stand and stand-to-sit activities favour a dense trabecular network over a thick cortex, while the opposite is true for walking or stair climbing activities. The number of shell elements with increased thickness can also explain the very small change of -0.4% (Fig 1) in cortical bone volume observed in this scenario. However, in scenarios $Femur_{NoStairs}$ and $Femur_{WalkOnly}$, there is a much greater proportion of elements resorbing than growing, while the total reduction in cortical bone volume remains <5%. This is likely due to the initial thickness of most of these thinned shells. Indeed, as shown in Figs 3 to 5, they are located at the extremities of the femur, or on the anterior and posterior faces of the shaft. As discussed in Phillips et al. [12], the cortex in

the original femur model is very thin at the extremities, consistent with clinical images, but also in the sagittal plane, which was identified as a limitation to this model. For this reason, a change of less than 50% of their thickness will not impact greatly the total cortical volume.

As can be seen on Fig 1, trabecular bone exhibits more volumic change than cortical bone in all three remodelling scenarios. This would point towards trabecular bone as a domain more sensitive to structural adaptation. There is an interesting parallel to draw here with the remodelling occurring in osteoportic conditions, where it is reported that two thirds of the bone loss occurs in trabecular bone against one third in the cortex [5]. It is conceivable that trabecular bone has a greater potential for adaptation to local heterogeneity of strains while cortical bone, which forms the outer shell of the long bone, would respond to heterogeneity of a greater scale, related to overall shape deformations. As shown in Phillips et al. [12], the loading associated with walking activity is complex enough, with high enough strain amplitudes, to generate strain field histories leading to a fairly 'complete' adaptation when compared to the 'optimal' adaptation to a combination of activities, with the exception of the trabecular architecture of the distal femur. This can also be inferred from Fig 5 where it is still possible to ascertain the presence of the main trabecular groups when the only activity left is walking, but shows a virtually resorbed distal trabecular lattice. The complexity of the loading scenario associated with walking might explain why the cortex is not required to adapt much in the activity regimes presented in this study. The literature reports a protective effect of walking [5] which supports this hypothesis. It might however be drastically different if the individual were to change their walking pattern (slower, wider stance, etc) or frequency of performance, as is sometimes reported in the elderly [24]. Fig 1 shows that the change in trabecular volume when stopping both stair climbing and sitting activities is roughly equal to the sum of the changes in trabecular volume when stopping only one of these activities. Based on this observation, it can be inferred that stair climbing and sitting activities are complementary activities in terms of trabecular adaptation: they target different areas. This statement is supported by the observations made on Figs 3 to 5 where the areas of trabecular bone resorption show very limited overlap in $Femur_{NoStairs}$ and $Femur_{NoSit}$ and are unified in $Femur_{WalkOnly}$. Based on the relatively large difference between the change in cortical volume when stopping both stair climbing and sitting activities and the sum of the changes when stopping only one activity, it can be hypothesized that stair climbing and sitting activities require the adaptation of partly overlapping zones, such that cortical bone is prevented from resorbing when only one activity is stopped, as the other is still mechanically stimulating these areas. Figs 3 and 4 would indicate otherwise, as their respective zones of bone resorption are distinct. A potential explanation would be that both activities target similar zones of the cortex, but to a different extent, meaning that the area would resorb when its primary stimulating activity is stopped, but only to the extent of the difference in mechanical stimulus between the two activities.

All four failure loads in side fall scenario are within the range reported for in-vitro testing in literature [19, 25, 26]. The intertrochanteric fracture patterns observed are consistent with typical patterns reported in literature [21, 25, 26]. The increased damage observed in the superior lateral aspect of the neck for $Femur_{NoSit}$ and the altered direction of the fracture line observed for $Femur_{WalkOnly}$ are consistent with the localised thinning of the cortex observed in these femurs compared to $Femur_{Original}$. On the other hand, the overall thinning of the trabecular lattice observed in the greater trochanter of $Femur_{NoStairs}$ or in the neck of $Femur_{NoSit}$ does not seem to greatly impact the fracture response. This would suggest a greater influence of cortical morphology on fracture behaviour compared to trabecular structure. A cessation of stair climbing and sitting activities leads to respectively 7% and 10% decrease in failure load. A cessation of both types of activities leads to a 21% decrease in failure load. In addition, a decrease in the load corresponding to the first crack appearance is observed following a

cessation of sit-to-stand and stand-to-sit activities, or to a higher extent when the activity regime is reduced to walking only. These findings are in broad agreement with studies investigating the impact of different activities on the strain experienced in the superior cortex of the femoral neck, although sit-to-stand and stand-to-sit activities were not included in the range of activities examined [27, 28]. These studies recommend stair ambulation and fast walking to increase the strain in the superior cortex with the aim of reducing fracture risk.

Limitations to this study are the arbitrary and binary aspects of the activities removed from the loading regimes: only sit-to-stand and stand-to-sit, and stair climbing and descending were tested, and only in two states: either performed or not. It is however likely that before stopping completely an activity, individuals would progressively modify their patterns. They would for instance start relying on hand rails, or pushing with their arms to stand up. Their walking habits might change as well in pace, stride length and stride width. Another limitation is the nature of the bone adaptation allowed in the model: only inner architecture, through changes in cross-sectional areas of elements, is subject to adaptation. The outer shape of the bone remains the same, while its adaptation, for example an increase in outer radius of the shaft, has also been observed in some clinical studies [29].

It is interesting to note that the physical activities involved in the daily routine of a young healthy adult are thought to reduce fracture risks by improving muscle strength and flexibility, balance and reaction time, and thus decreasing the risk of falling itself, as well as the force associated with the impact [2]. Overall, this study yielded strong indications that this type of physical activity also provides a substantial protection against bone loss, thereby maintaining a higher fracture load under side fall.

## Supporting information

**S1 File.**
(INP)

**S2 File.**
(INP)

**S3 File.**
(INP)

**S4 File.**
(INP)

## Author Contributions

**Conceptualization:** Claire C. Villette, Andrew T. M. Phillips.

**Data curation:** Claire C. Villette, Andrew T. M. Phillips.

**Formal analysis:** Claire C. Villette, Andrew T. M. Phillips.

**Investigation:** Claire C. Villette, Andrew T. M. Phillips.

**Methodology:** Claire C. Villette, Andrew T. M. Phillips.

**Supervision:** Andrew T. M. Phillips.

**Validation:** Claire C. Villette, Andrew T. M. Phillips.

**Visualization:** Claire C. Villette, Andrew T. M. Phillips.

**Writing – original draft:** Claire C. Villette.

**Writing – review & editing:** Claire C. Villette, Andrew T. M. Phillips.

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
