## [Decision Letter · Decision Letter 0]

9 Oct 2023

PONE-D-23-29003Influence of a change in activity regime on femoral bone architecture and failure behaviourPLOS ONE

Dear Dr. Phillips,

Thank you for submitting your manuscript to PLOS ONE. After careful consideration, we feel that it has merit but does not fully meet PLOS ONE’s publication criteria as it currently stands. Therefore, we invite you to submit a revised version of the manuscript that addresses the points raised during the review process.

We look forward to receiving your revised manuscript.

Kind regards,

Ewa Tomaszewska, DVM Ph.D

Academic Editor

PLOS ONE

Journal Requirements:

3. Thank you for stating the following financial disclosure: "CV was funded by the Royal British Legion Centre for Blast Injury Studies at Imperial College London".

Reviewers' comments:

Reviewer's Responses to Questions

**Comments to the Author**

1. Is the manuscript technically sound, and do the data support the conclusions?

Reviewer #1: Yes

2. Has the statistical analysis been performed appropriately and rigorously? 

Reviewer #1: N/A

3. Have the authors made all data underlying the findings in their manuscript fully available?

Reviewer #1: Yes

4. Is the manuscript presented in an intelligible fashion and written in standard English?

Reviewer #1: Yes

5. Review Comments to the Author

Reviewer #1: The manuscript is generally well-written and presents a thoughtfully designed experiment. However, there are minor issues that the authors need to address.

I recommend introducing the scientific (medical) nomenclature for femur sections: diaphysis, metaphysis, and epiphysis. While these terms don't need to be used continuously throughout the manuscript, they are crucial, especially given the topic of bone fractures. The location of the fracture on the bone—whether on the diaphysis, metaphysis, or epiphysis—often determines the severity, implications, and health consequences.

L29: The abbreviation "FE" should be explained upon its first usage.

L33-34: Mechanostat is more appropriately described as a hypothesis or model rather than a principle.

L77-78: For better readability, consider revising to: “Yield strain (y) in compression (c) and tension (t), and ultimate strains (u)…”

L79: Please replace “y” with “c” in the ultimate tension formula.

L83: Use lowercase for "modulus".

L84: For clarity, consider writing as: E = 18.0 GPa.

L95, other results, and Fig. 1: To enhance clarity, consider adding a decimal separator and a zero (".0") to integer values.

L105-6: Please refer to these as the distal and proximal halves of the bone, indicating the respective diaphysis, metaphysis, and epiphysis.

L116 and subsequent lines: "Diaphysis" is the precise anatomical term, preferable over "shaft". If the target audience includes technical readers such as engineers, they might find "shaft" more familiar. However, "diaphysis" remains the scientifically accurate term.

L122 and other lines: The term "bone resorption" is medical in nature. Given your focus on FE modeling, I suggest a rephrase like: "Most of the distal lateral trabecular bone exhibits changes (reductions) that may be interpreted as bone resorption...". Please consider implementing this correction in both the results and discussion sections, specifically the first instance it appears in each section. As I am not a native English speaker, I trust you can find a more optimal and appropriate way to phrase this if you choose to make the suggested changes.

L148 and L153: Please amend to “Steen Jensen and Michaelsen’s classification”. Also, consider citing the original works available online 10.3109/17453677508989266 or 10.3109/17453678008990877)

L177: Correct to “delta_t” (without the h subscript).

Tables 1 and 2: In the penultimate columns, please correct to the interval “[50,90]”. Clarify if presented intervals are left- or right-open.

Fig 1: Ensure figure captions stand alone in meaning. Add the reference to the original work by Phillips et al. and correct to “the original femur model [17] and each of the lower intensity activity regimes”

Fig 3-5: Amend to “c) the proximal and d) the distal epiphysis in … “

Fig. 7: The caption for the y-axis appears to be missing.

6. PLOS authors have the option to publish the peer review history of their article (what does this mean?). If published, this will include your full peer review and any attached files.

Reviewer #1: No

---

## [Author Response · Author response to Decision Letter 0]

10 Jan 2024

We have included the response to the reviewer and editor comments in the cover letter.

---

## [Decision Letter · Decision Letter 1]

16 Jan 2024

Influence of a change in activity regime on femoral bone architecture and failure behaviour

PONE-D-23-29003R1

Dear Dr. Andrew T. M. Phillips,

We’re pleased to inform you that your manuscript has been judged scientifically suitable for publication and will be formally accepted for publication once it meets all outstanding technical requirements.

Kind regards,

Ewa Tomaszewska, DVM Ph.D

Academic Editor

PLOS ONE

Additional Editor Comments (optional):

Reviewers' comments:

Reviewer's Responses to Questions

**Comments to the Author**

1. If the authors have adequately addressed your comments raised in a previous round of review and you feel that this manuscript is now acceptable for publication, you may indicate that here to bypass the “Comments to the Author” section, enter your conflict of interest statement in the “Confidential to Editor” section, and submit your "Accept" recommendation.

Reviewer #1: (No Response)

2. Is the manuscript technically sound, and do the data support the conclusions?

Reviewer #1: Yes

3. Has the statistical analysis been performed appropriately and rigorously? 

Reviewer #1: Yes

4. Have the authors made all data underlying the findings in their manuscript fully available?

Reviewer #1: Yes

5. Is the manuscript presented in an intelligible fashion and written in standard English?

Reviewer #1: Yes

6. Review Comments to the Author

Reviewer #1: I would like to thank the authors for reviewing the comments and suggestions. In my opinion the article is now acceptable for publication.

7. PLOS authors have the option to publish the peer review history of their article (what does this mean?). If published, this will include your full peer review and any attached files.

Reviewer #1: No
